# Urban Sprawl and Carbon Emissions Effects in City Areas Based on System Dynamics: A Case Study of Changsha City

**Luyun Liu [1,2], Yanli Tang [3,\*], Yuanyuan Chen [1], Xu Zhou [1,2] and Komi Bernard Bedra [4]**

1   School of Landscape Architecture, Central South University of Forestry and Technology, Changsha 410004, China; t20172369@csuft.edu.cn (L.L.); 20201100304@csuft.edu.cn (Y.C.); t20080238@csuft.edu.cn (X.Z.)
2   Hunan Big Data Engineering Technology Research Center of Natural Protected Areas Landscape Resource, Changsha 410004, China
3   Department of Architecture and Art, Changsha University of Science and Technology, Changsha 410076, China
4   School of Architecture and Art, Central South University, Changsha 410083, China; komibedra@csu.edu.cn
\*   Correspondence: 008207@stu.csust.edu.cn; Tel.: +86-1511-6345-886

**Abstract:** Climate change is a global problem facing mankind, and achieving peak $CO_2$ emissions and carbon neutrality is an important task for China to respond to global climate change. The quantitative evaluation of the trends of urban energy consumption and carbon emissions is a premise for achieving this goal. Therefore, from the perspective of urban expansion, this paper analyzes the complex relationship between the mutual interactions and feedback between urban population, land expansion, economic growth, energy structure and carbon emissions. STELLA simulation software is used to establish a system dynamics model of urban-level carbon emissions effects, and Changsha city is used for the case study. The simulated outputs of energy consumption and carbon emissions cover the period from 1949 to 2016. From 1949 to 2016, Changsha's total energy consumption and carbon emissions per capita have continuously grown. The total carbon emissions increased from 0.66 Mt-$CO_2$ to 60.95 Mt-$CO_2$, while the per capita carbon emissions increased from 1.73 t-$CO_2$/10,000 people to 18.3 Mt-$CO_2$/10,000 people. The analysis of the structure of carbon emissions shows that the industrial sector accounted for the largest proportion of emissions, but it had gradually dropped from between 60% and 70% to about 40%. The carbon emissions of residential and commercial services accounted for less than 25%, and the proportion of transportation carbon emissions fluctuated greatly in 2013 and 2016. From the perspective of carbon emissions effects, carbon emissions per unit of GDP had a clear downward trend, from 186.11 t-$CO_2$/CNY$10^4$ to 1.33 t-$CO_2$/CNY$10^4$, and carbon emissions per unit of land showed two inflection points: one in 1961 and the other in 1996. The general trend showed an increase first, followed by a decrease, then a stabilization. There is a certain linear correlation between the compactness of urban shape and the overall trend of carbon emissions intensity, while the urban shape index has no linear correlation with the growth rate of carbon emissions. The carbon emissions assessment model constructed in this paper can be used by other municipalities, and the assessment results can provide guidance for future energy planning and decision making.

**Keywords:** urban sprawl; carbon emission; system dynamics model; STELLA

## 1. Introduction

Climate change is a global problem facing mankind. China's strong commitment to addressing global climate change is to reach "peak $CO_2$ emissions by 2030 and achieve carbon neutrality by 2060". The acceleration of urbanization and industrialization in the 20th and 21st centuries has greatly changed the land use and cover of modern cities, resulting in the degradation of urban microclimates and the rise of urban air temperatures [1]. Up to 80% of greenhouse gas (GHG) emissions are generated by more than half of the population

living in urban areas [2]. The quantitative assessment of carbon emission trends and their emission effects based on urban expansion is the basis and prerequisite work for achieving the goal of "carbon peak and carbon neutrality". This research is particularly important.

In recent years, there has been a wide variety of literature about quantitative research on urban carbon emissions. The literature covers different areas such as space layout, transportation mode, resource utilization, economic development, technology application, lifestyle and other related disciplines. In terms of modeling scale, the literature covers the macro scale (national and regional) and the meso scale (urban). In relation to urban and rural planning, more attention has been paid to the planning of the physical space. At the national level, for instance, Genice, K., using Mexico as an example, proposed eight energy efficiency measures with regard to energy use in the residential, commercial and public sectors [3]. Flavio, R. describes the results of a study of Ecuador's energy status, which used the system dynamics methodology to model supply, demand and $CO_2$ emissions scenarios for the year 2030 [4]. Taking Hong Kong as a case study, Geoffery, S.Q. predicted the sustainability of urban land use and suggested development strategies for cities with different population densities [5]. Sun, W. established a carbon emission prediction model based on the short-term prediction of carbon emissions and the combined model of "decomposition-prediction" [6]. Li, Y. put forward the KLS (Kalman filtering, long short-term memory, support vector machine) model based on a novel time-series prediction method to forecast Chinese carbon emissions [7]. Kabindra Adhikari proposed a space-for-time substitution model to predict current and future soil organic carbon (SOC) stocks in Wisconsin [8]. At the regional level, Liu, Y. took the carbon emission panel data of six provinces in Central China as basic data and used the LMDI decomposition method to study the temporal and spatial evolution characteristics and convergence of carbon emissions [9]. Based on the night light data of Beijing, Tianjin and Hebei from 1992 to 2012, Su, X. estimated the carbon emissions of Beijing, Tianjin and Hebei by using an ArcGIS spatial statistics and spatial econometric model, and analyzed its temporal and spatial evolution characteristics and influencing factors [10]. Imran Hanif employed the Environmental Kuznets Curve (EKC) hypothesis in studying the impact of energy consumption, economic growth, and urbanization on carbon emissions in developing economies [11]. With figures showing that approximately 40% of the UK's carbon emissions are attributable to household and transport behaviors, policy initiatives have progressively focused on the facilitation of "sustainable behaviours" [12]. At these two scales, the existing literature mainly focuses on establishing prediction models based the relationship between regional energy consumption, carbon emission trends and urbanization (population density, spatial expansion, economic growth, etc.). At the municipal level, four main aspects have been considered: low-carbon construction and urban development, carbon emission and urban spatial structure, carbon emission and urban form, and carbon emission and urban sprawl. For instance, Pan, H.X. analyzed the methodology of urban planning formulation at the region, city and street block levels, and several arguments on urban land use, transport, density control and mixture of land use have been discussed regarding a low carbon urban spatial strategy [13]. Week and Hiroshi established the basic database of urban carbon emissions in Iskandar, Malaysia, and used system dynamics modeling technology to predict the trend of urban energy consumption and carbon emissions [14,15]. Based on night light data, Chen, J.D. forecasted the urban carbon dioxide emissions of 334 prefecture level cities in China from 1992 to 2017 and put forward carbon reduction strategies [16]. Feng, Y.Y. and Yuan, Q.M. studied the trend characteristics and influencing factors of carbon emissions in Beijing and Tianjin, respectively [17,18]. Tong, H.F. focused on the four modules of population, economic production, energy and water resources, and constructed a dynamic model of sustainable development in Beijing [19]. Wu, M. modeled and analyzed the carbon emissions effect of land use under different policy scenarios in Wuhan from the impact of land, population, society, economy and energy on carbon emissions [20]. Xu, J. modeled from the perspective of landscape ecological risk and environmental pollution and compared the landscape risk assessment of four urban layout

modes [21]. Sun, Y.H. constructed an urban sustainable development evaluation model covering five subsystems: economy, population, water resources, environment, and science, technology and education [22]. Guo, J. points out that the key to urban development transformation is the coordination of the industrial structure and optimization of the spatial structure to achieve low carbon goals [23]. A GIS-based model was proposed for testing how urban form and building typology affect energy performance and carbon emissions in the City of Macau [24]. Ewing, R. suggested that compact urban spaces can effectively reduce the energy consumption of the urban residents, and thus reduce carbon dioxide emissions [25]. Chen, Z.Q. used the regression model to test the impact of the urban form on carbon emissions, and he also put forward a low carbon development framework of "spatial-land-transportation" to reduce the level of urban carbon emissions [26]. Carpio, A. analyzed the urban expansion of the Monterrey Metropolitan Area (MMA) in Mexico from 1990 to 2019 using satellite imagery and geographic information systems (GIS) to determine its relation to carbon emissions [27]. Hong, Y. indicated that designing cities to be compact with less accessibility to green spaces (GS) and water bodies (WB) may increase household energy consumption. The substantial use of GS and WB, even if fragmented, will reduce carbon emissions of residential energy use [28]. Li, J.J. analyzed the relationship between land use and carbon emission intensity in Shanghai. The study specifically established the relationship between urban compactness and industrial land use using three models: a space compactness model, an energy consumption model and a carbon emission intensity model [29]. The experiments conducted by Shao, Z.F [30] showed that urban sprawl resulted in unsustainable urban development patterns from the social, environmental, and economic perspectives. Zhang, M. calculated the levels of land intensive use and land use carbon emissions from 1996 to 2010 in three central cities in Hubei Province, and the results showed that there is a long-term equilibrium between intensive land use level and land use carbon emissions [31]. Research at the urban level has mainly focused on new towns. The concept of a new town in China generally refers to newer urban development areas. Liu L.Y. put forward the dynamic model framework of a new town carbon emission system from the four aspects of construction, industry, transportation and natural carbon sink, and compared the carbon emission effects of the new town spatial planning schemes under different policy scenarios [32,33].

Previous studies have conducted multi-scale, quantitative modeling of carbon emissions at national, provincial, municipal or county levels. The quantitative methods used have included the logarithmic mean Divisia index method (LMDI), geographically weighted regression (GWR) model, time series prediction and variables regression, system dynamics method, etc. Research conducted from the perspective of urban spatial expansion has mostly considered two aspects: night light data and the dynamic transfer change of land use. In fact, a land use map of a given city can accurately reflect the changes that occurred during that city's spatial expansion. The influencing factors mainly include land, population, economy, energy, urban layout modes, urban form, patterns and neighborhood design (e.g., urban building typology and assemblage). The research also shows that the system dynamics method is a better method for studying urban carbon emissions. Even though it has been applied, to a certain extent, to the prediction of carbon emission trends in cities and new towns, its application at the urban scale is still very rare. Based on the recent 60 years of energy consumption data and on present land use, this paper constructs a dynamic model considering five urban operation subsystems: residential, commercial, industrial, transportation and green space carbon sink. The data on energy consumption were obtained from the energy statistical yearbook from 1949 to 2017, and the land use data directly from the land use map of the current urban master plan. The paper simulates the dynamic change index of the total carbon emissions, the carbon emissions intensity, the carbon emissions structure, the carbon emissions per unit land, and summarizes the time-rules and the contribution features to carbon emissions. This paper intends to provide the necessary guidance and suggestions for future energy planning and decision making.

The carbon emissions assessment model constructed in this paper can be used by other municipal cities.

## 2. Study Area and Methods

### 2.1. Overview of the Study Area

Located in central China, Changsha is defined as an important city in the central region in the national development strategy. By July 2021, the population scale of Changsha's central urban area had exceeded 5 million, which is one of 17 mega cities in China (the population scale of central urban area is between 5–10 million, which is a mega city). Like other metropolises in developing countries, Changsha is undergoing rapid development and transformation, with an accelerated industrialization and urbanization, and from 1949 to 2016, the urban population of Changsha increased from 383,500 to 3.36 million, the regional GDP from CNY 2.87 million to CNY 8510.13 million, and the built-up area from 6.7 km$^2$ to 476.34 km$^2$. During this period, the city's master plan went through six revisions in 1979, 1996, 2003, 2008, 2013 and 2016, respectively. The respective land use maps are shown in Figure 1, the urban land areas are shown in Table 1 and the urban sprawl intensities of various urban land areas are shown in Table 2. Figure 1 and Table 1 show that urban construction land is increasing, agricultural and forestry land are decreasing, urban compactness shows a downward trend and is "spread out like a pie", and the urban shape index has had little change. The expansion intensity of residential land is relatively the largest, especially after 2008, as it is several times that of other types of land. The expansion intensity of commercial service land is relatively stable. The expansion intensity of industrial land continues to decline, and the expansion intensity of road traffic land is relatively stable. Changsha is an energy-deficient city with an insufficient capacity to supply for a continuously growing energy demand. 80% of its energy needs to be constantly adjusted. The self-produced energy capacity is weak, and the growth potential is limited. The potential for new energies is insufficient, and the carbon emissions effect from energy consumption is very likely to grow in the future. How to balance the carbon emissions caused by rapid urban expansion and the serious energy-deficiency are important issues that need to be addressed.

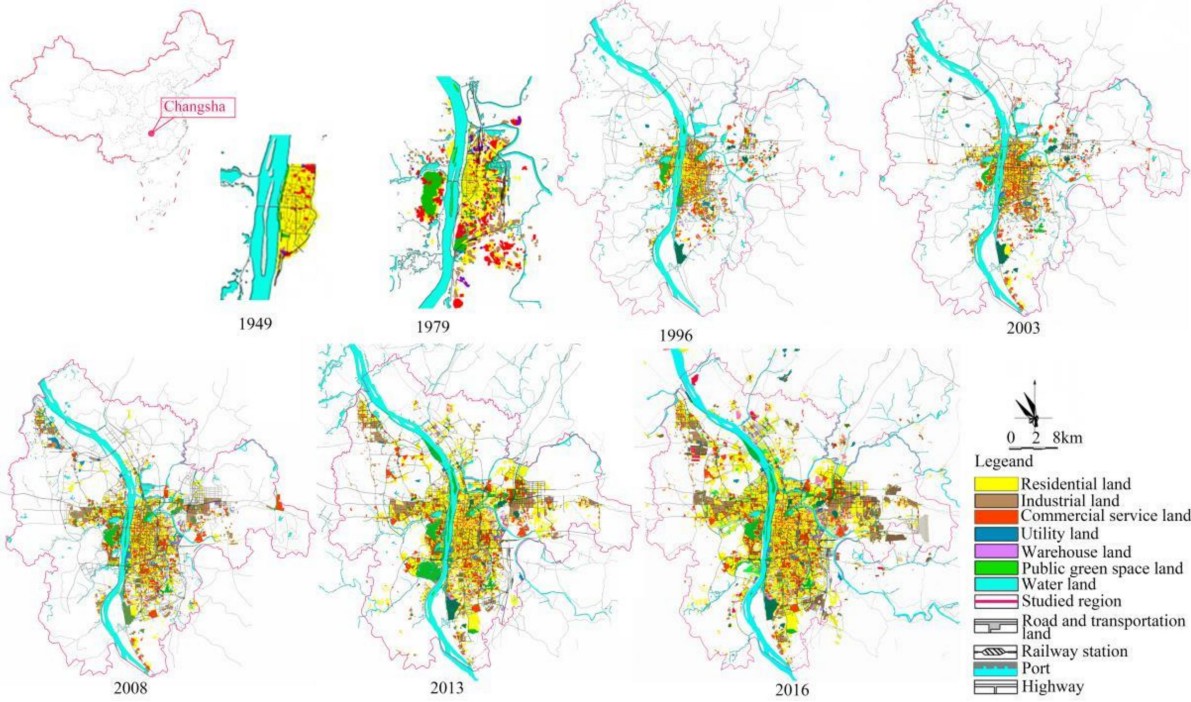

**Figure 1.** Land use change from 1949 to 2016.

**Table 1.** Urban land use indices, urban compactness, and urban morphology indices.

| | Item | 1949 | 1979 | 1996 | 2003 | 2008 | 2013 | 2016 |
|---|---|---|---|---|---|---|---|---|
| | The built-up area | 6.7 | 51.8 | 104.93 | 96.26 | 209.63 | 393.78 | 476.34 |
| | Residential land (R) | 1.92 | 11.02 | 25.38 | 30.95 | 69.95 | 134.79 | 155.32 |
| | Commercial service land (A and B) | 1.26 | 13.33 | 26.97 | 37.02 | 43.98 | 73.4 | 66.58 |
| | Industrial land (M) | 1.52 | 12.5 | 21.9 | 27.1 | 30.6 | 66.5 | 94.56 |
| Urban land area | Road and transportation land (S) | 0.8 | 2.97 | 9.59 | 20.58 | 31.76 | 37.67 | 52.87 |
| | Public green space (G) | 0.6 | 4.8 | 7.94 | 12.29 | 21.42 | 29.13 | 34.13 |
| | Waters (E1) | 25.26 | 73.45 | 82.3 | 85.6 | 85.6 | 93 | 95.65 |
| | Arable land (E21) | 193.89 | 177.83 | 151.73 | 143.61 | 104.68 | 73.5 | 52.3 |
| | Garden land (E22) | 179.38 | 164.53 | 140.38 | 132.87 | 96.84 | 68 | 35.2 |
| | Forest land (E23) | 96.02 | 88.07 | 75.14 | 71.12 | 51.84 | 36.4 | 24.3 |
| Urban compactness (U) [1] | | 0.521 | 0.228 | 0.172 | 0.146 | 0.21 | 0.205 | 0.195 |
| Urban morphology indices(I) [2] | | 34.0 | 28.81 | 33.80 | 33.29 | 31.89 | 26.17 | 25.64 |

Note: urban land use area contains the eight categories of R, A, B, M, W, S, U, and G, codes for the classification of urban land which refer to GB50137-2011. W (warehouse land) and U (utility land) have little to do with urban carbon emissions. Therefore, classes W and U are not described in the table. The unit area for each type of urban land is sq. km. [1] ref. [34] proposed that the methodology for measuring compactness be adopted based on the urban land use GIS raster analysis and resorting to the gravitation approach. [2] The urban morphology indices are obtained by calculating the ratio of the standard circular shape to the urban boundary shape [35]. For the urban morphology indices, $(I) = \Sigma n_{i=1} | (R_i / \Sigma n_{i=1} R_i \times 100 - 100/n) |$, where r is the radial distance as measured from the centroid of the built-up area to its edge, and *n* is the number of regularly spaced radii.

**Table 2.** Urban sprawl intensity [1].

| Item | 1949–1979 | 1979–1996 | 1996–2003 | 2003–2008 | 2008–2013 | 2003–2016 |
|---|---|---|---|---|---|---|
| Residential land (R) | 28.66 | 21.27 | 24.19 | 19.29 | 33.37 | 34.23 |
| Commercial service land (A and B) | 18.81 | 25.73 | 25.7 | 23.08 | 20.98 | 14.52 |
| Industrial land (M) | 22.69 | 24.13 | 20.87 | 16.89 | 14.6 | 16.89 |
| Road and transportation land (S) | 11.94 | 5.73 | 9.14 | 12.83 | 15.15 | 9.57 |
| Public green space (G) | 8.96 | 9.27 | 7.57 | 7.66 | 10.22 | 7.4 |

[1] Urban sprawl intensity reflects the change in the rate of land use expansion in the study area, which is monitored by the urbanization intensity index mode [36]. $U = (U_b - U_a)/A \times T$, where U denotes the urban sprawl intensity index; $U_b$ and $U_a$ are each type of urban land area at the final and initial stages, respectively; A is the total of urban land use area; and T is the period of time.

First of all, Changsha is one of the most representative cities that is experiencing rapid urban expansion and economic development, and the carbon emissions of energy consumption are increasing. Secondly, the data on energy consumption over the years in the urban statistical yearbook is relatively complete. Third, we have six consecutive urban land use data sets provided by the Changsha Survey and Design Institute. Based on the above reasons, we selected Changsha City as the case study area.

*2.2. System Dynamics Modeling Theory*

System dynamics takes computer simulation technology as the main method to solve problems in complex dynamic feedback systems through structural function analysis [37]. It was originally developed by Professor Forrester of the Massachusetts Institute of Technology in 1956 and was widely used in the field of social sciences. In 1972, Meadows D.'s research, "The Limits of Growth" [38], which reported on human environmental issues, triggered a new concept of global sustainable development. Since system dynamics has absorbed the essence of cybernetics and information theory on the basis of system theory, it has been studying real, complex problems through structural function analysis and

information feedback. In recent years, many scholars have used system dynamics methods to study urban environmental problems.

In urban energy management, system dynamics is mainly applied to the following aspects: Wee, K.F. specifically considered the future $CO_2$ emissions trend and adopted an FML model as a decision-making tool in structural planning [15]. In order to estimate and predict the trend of urban energy consumption and carbon dioxide emissions in Beijing, Feng and Zhang developed the Beijing-STELLA model [39]. From the perspective of regional typology, Li, Z.H. established a system dynamics model called EECP (economy energy carbon emissions policy) to simulate the trend of carbon dioxide emissions [40]. STELLA, Ithinks and Vensim have been widely used in previous studies. After analysis, STELLA expresses the causal relationship between various factors more clearly and intuitively. Therefore, STELLA is used in this study to simulate urban energy consumption and related carbon emissions.

## 3. System Dynamic Model for Urban Energy Consumption and $CO_2$ Emissions

### 3.1. Overall Framework of the Model

According to the modeling method of system dynamics, the modeling goal is to evaluate the impact of energy consumption on carbon emissions by analyzing the input, operation and output characteristics of urban carbon emissions system, and then establishing a causal feedback loop diagram. Therefore, the overall framework of the carbon emissions effect model of urban energy consumption takes population growth and energy consumption as the basic factors, and considers the relationship between industrial structure, economic growth and land expansion. The model defines five corresponding sub-models, namely, a residential sub-model ($F_{Res}$), a commercial service sub-model ($F_{Com}$), an industrial sub-model ($F_{Ind}$), a transportation sub-model ($F_{Tra}$) and a carbon sequestration sub-model ($F_{Ga}$). The overall frame and the feedback relationships among the factors are shown in Figure 2. In the model, the inputs include the consumption of fossil energies (e.g., coal, liquefied petroleum gas (LPG), gas, natural gas (NG) and electricity) through the five urban subsystems: residence, commercial services, industrial, transportation and carbon sink. The mutual conversions of materials consumption and energy between the subsystems are analyzed, and, finally, converted into carbon dioxide emissions. The outputs include carbon emissions and indices (such as the total $CO_2$ emissions, the emissions intensity, the emissions per unit land, and the per capita emissions). The boundary of the model is the city's perimeter, and the period of concern for the dynamic model is from 1949 to 2016.

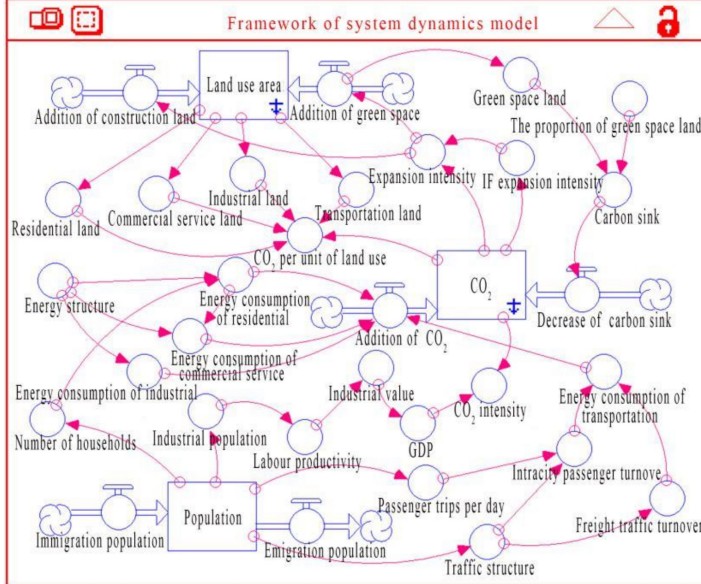

**Figure 2.** A causal feedback loop diagram for the urban $CO_2$ emissions model.

### 3.2. Residential Sub-Model

The carbon emissions of the residential sub-model mainly depend on the total urban population and are determined by the number of households per capita, the electricity consumption per capita, the coal consumption per capita, the gas consumption of each household, the proportion of liquefied petroleum gas and natural gas, and the various types of energy consumption per unit. The carbon emissions per unit area of the residential sector is also related to the intensity of the residential land expansion. The greater the land expansion intensity is within a certain period of time with the total carbon emissions remaining unchanged, the smaller the carbon emissions intensity per unit of land. The energy intensity ($CO_2$ emissions intensity per unit GDP) also has a certain constraint on the urban land expansion intensity. The greater the total carbon emissions, the higher the economic output, the higher the benefit, and the smaller the intensity of carbon emissions, which means that the higher the input–output per unit of land, the more economical the land use is. The stock and. flow diagram for the residential sub-model is shown in Figure 3a and the formulation of the residential sub-model is as shown as Equation (1):

$$F_{\text{Res}} = H \times \sum (CnR_i \times EmR_i) \tag{1}$$

where $F_{\text{Res}}$ is the $CO_2$ emissions of the residential sector, $H$ is the total number of urban households, $CnR$ is the energy consumption of $i$-type in the residential sector, $EmR$ is the carbon emissions per unit of energy consumption and $i$ is the energy type, which refers to coal, liquefied petroleum gas (LPG), gas, natural gas (NG) or electricity in the residential sector.

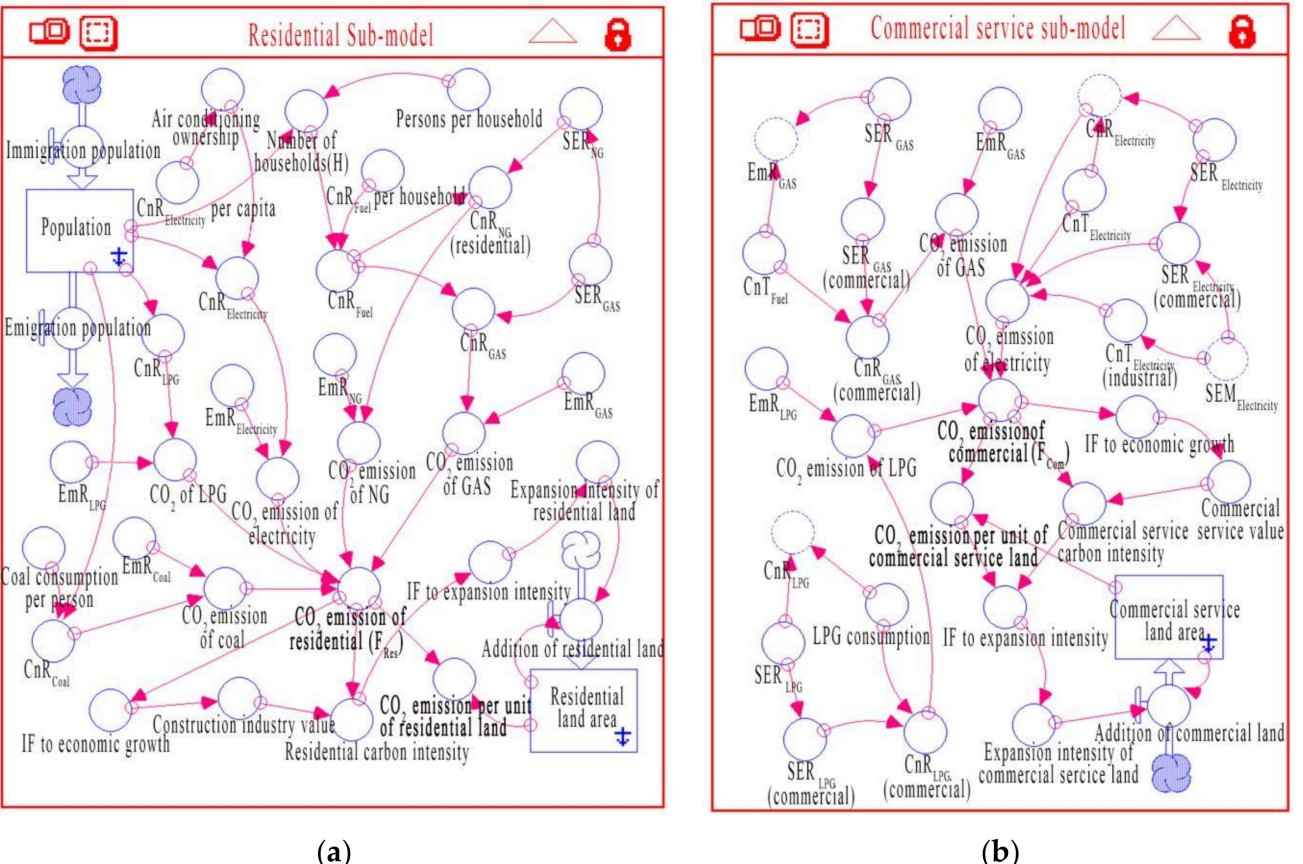

(**a**)          (**b**)

**Figure 3.** The $CO_2$ emissions stock and flow diagram. (**a**) Residential sub-model. (**b**) Commercial service sub-model.

### 3.3. Commercial Service Sub-Model

The carbon emissions of the commercial service sub-model mainly refer to the carbon emissions generated by the indirect energy consumption required by the urban permanent population for daily life and work (the direct energy consumption is attributed to the residential sub-model). These indirect carbon emissions are caused by the consumption of energy in the production and supply of non-energy goods and services, such as clothing, food, housing and transportation in the daily life of residents. The energy consumption is also consistent with the energy consumption data in the statistical yearbook of Changsha over the years. The carbon emissions from energy consumption are mainly estimated considering the consumption of liquefied petroleum gas, pipeline gas and electricity, which are converted into the consumption of standard coal, and then converted into urban $CO_2$ emissions. The commercial energy consumption can be obtained from the total energy consumption minus the industrial energy consumption for living and the direct energy consumption for residents' daily lives. The stock and flow diagram for the commercial service sub-model is shown in Figure 3b and the formulation of the residential sub-model is as shown in Equation (2):

$$F_{\text{Com}} = \sum [(CnT_i \times EmR_i) - (SER_i \times CnT_i \times EmR_i) - (SEM_i \times CnT_i \times EmR_i)] \quad (2)$$

where $F_{\text{Com}}$ is the carbon emissions in the commercial service sector, $CnT$ is the total energy consumption of $i$-type energy, $SER$ is the share of $i$-type energy consumption directly used for daily life, $SEM$ is the share of $i$-type industrial energy used in the living part, $EmR$ is the carbon emissions per unit of energy consumption and $i$ is the energy type, which refers to liquefied petroleum gas (LPG), gas, natural gas (NG) or electricity in the commercial service sector.

### 3.4. Industrial Sub-Model

The energy consumption of industrial carbon emissions sub-model mainly considers steel, non-ferrous metals and ferrous metals, and the building materials industries, chemical industries and petroleum processing, textile and machinery manufacturing industries. The energy consumption precisely includes raw coal, washed coals, other washed coals, coal products, coke, gas, natural gas and liquefied natural gas. For other washed coals, various energy sources, such as gasoline, kerosene, diesel, fuel oil, liquefied petroleum gas (LPG), other petroleum products, heat, electricity and other biofuels, are converted into standard coal and then converted into carbon dioxide emissions. The stock and flow diagram for the commercial service sub-model is shown in Figure 4a and the formulation of the industrial sub-model is as shown in Equation (3):

$$F_{\text{Ind}} = \sum (E_{\text{Ind}} \times SE_i \times EmR_i) \quad (3)$$

where $F_{\text{Ind}}$ is the carbon emissions in the industrial sector, $E_{\text{Ind}}$ is the total industrial sector energy consumption, $SE$ is the share of energy consumption, $EmR$ is the carbon emissions per unit of energy consumption and $i$ is the energy type, which refers to raw coal (F1), washed coal (F2), other washed coal (F3), coal products (F4), coke (F5), producer gas (F6), NG (F7), liquefied natural gas (F8), gasoline (F9), kerosene (F10), diesel (F11), fuel oil (F12), LPG (F13), other petroleum products (F14), heat (F15), electricity (F16) and other biofuels (F17) in the industrial sector.

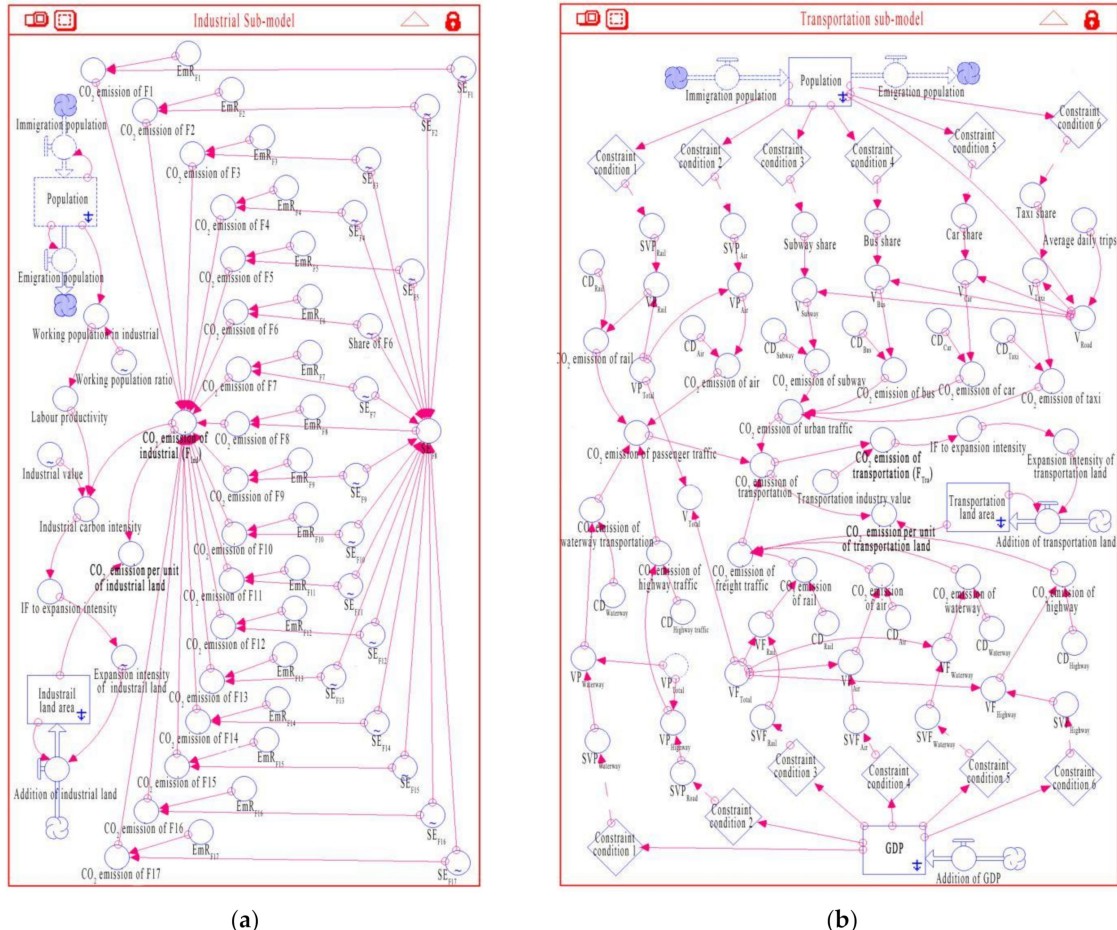

(**a**)   (**b**)

**Figure 4.** The CO₂ emissions stock and flow diagram. (**a**) Industrial sub-model. (**b**) Transportation sub-Model.

### 3.5. Transportation Sub-Model

The transportation sub model mainly includes direct carbon emissions and indirect carbon emissions. The urban scale transportation energy consumption mainly includes direct carbon emissions; that is, the sum of external traffic carbon emissions generated by the passenger transport turnover and freight turnover of railways highways, water transportation, aviation and pipeline transportation, and the internal traffic carbon emissions generated by urban public buses (electric vehicles) and taxis. Indirect emissions refer to the parts of carbon emissions produced by industrial development on energy consumption, and are not listed in the statistical yearbook. Instead, indirect emissions are included in the corresponding production department. The stock and flow diagram for the residential sub-model is shown in Figure 4b and the formulation of the residential sub-model is as shown in Equation (4):

$$F_{\text{Tra}} = \sum (VP_{\text{Total}} \times SVP_i \times K_i + VF_{\text{Total}} \times SVF_i) \times EmT_i + \sum (V_{\text{Road}} \times V_i) \times D_i \times CD_i \quad (4)$$

where $F_{\text{Tra}}$ is the carbon emissions in transportation sector, $VP_{\text{Total}}$ is the passenger turnover, $VF_{\text{Total}}$ is the freight turnover, $K_i$ is the conversion coefficient, $SVP$ is the share of passenger turnover, $SVF$ is the share of freight turnover, $EmT$ represents carbon emissions per unit of energy consumption, $V_{\text{Road}}$ is the daily passenger demand in the city, $V$ is the share of daily passenger demand, I is the transport mode (i.e., air, subway, high traffic, waterway, rail, bus, car, taxi, etc.), $D$ is the daily travel distance and $CD$ is the CO₂ emissions per unit of mileage of the *i*-type of vehicle.

### 3.6. Carbon Sequestration Sub-Model

Carbon sequestration encapsulates both natural carbon sequestration and carbon removal. It refers to the flow of carbon from the atmosphere to the biosphere. Based on the

requirements in terms of data integrity and measurement accuracy, various approaches may be appropriate for the study of carbon sequestration. Two main factors are closely related to the amount of carbon sinks: the first is green spaces, including water areas, cultivated lands, garden lands, forest lands, grasslands, other non-developed land and public green lands, and the second is the carbon absorption coefficient per unit of green land. The stock and flow diagram for the residential sub-model is shown in Figure 5, and the formula for the residential sub-model is as shown in Equation (5):

$$F_{\text{Ga}} = \sum (G_{\text{Total}} \times S_i) \times \Phi_i \tag{5}$$

where $F_{\text{Ga}}$ is the carbon sink, $G_{\text{Total}}$ is the total area of green lands, $S$ is the share of the green land area, $\Phi$ is the carbon sequestration coefficient per green land unit [24] and $i$ is the land type, which refers to water (E1), arable land (E21), garden land (E22), woodland (E23), grassland (E24), other non-development land (E9) or public green space (G).

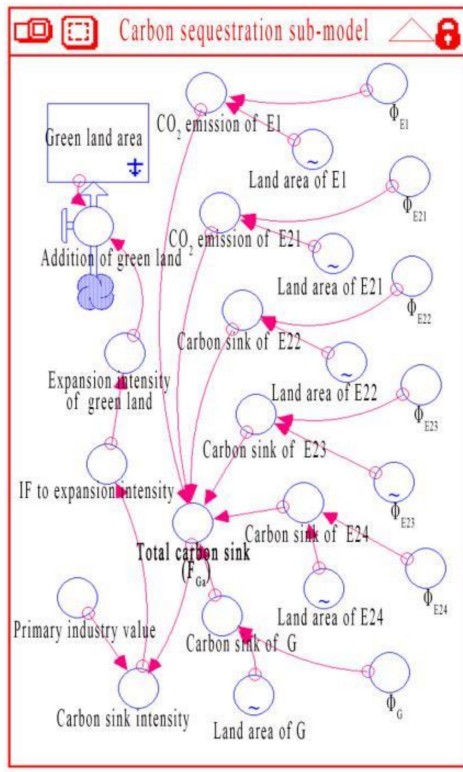

**Figure 5.** The $CO_2$ emissions stock and flow diagram of the carbon sequestration sub-model.

*3.7. Data Source and Description*

The parameters in the model include constant values, table functions, initial values, etc. The data sources are as follows:

- Statistical yearbooks. Some data can be directly extracted from the statistical yearbooks, such as the number of households, the capita per household, the urban permanent population and the GDP of Changsha from 1949 to 2016; the GDP outputs of the construction industry, the commercial service industry and the transportation, storage, postal, etc. industries; the freight transport volume, the freight turnover, the passenger transport volume and the passenger turnover; the proportions of railway, highway, water transport and civil aviation in the total turnover; and the residential, industrial and total fossil energy consumption data. The energy consumption of the commercial sector is obtained by subtracting the energy consumption of the residential and industrial sectors. Some data are obtained through the secondary processing of the originally available data, for example, the carbon emissions intensity index from

energy consumption in various industries can be calculated according to the total energy consumption and the industrial added value [41].

- Function fitting method. The time series method is used to perform regression and correlation analysis on the available data, and the fitting function determines parameter equations, such as the population growth rate, the GDP growth rate and other indicators.
- Table function method. For many nonlinear data relations, such as the carbon emissions intensities of various forms of energy consumption, the table function method is widely used in the model [42].
- Conditional function method. For known conditions that will change, and when the value of the function equation under different conditions is different, the expression of the conditional function is used in the model. For example, the traffic travel structure in the city will change with the change of the urban population, and the traffic travel ratio is expressed by the conditional function of the population size.

## 4. Results

### 4.1. Total Carbon Emissions and Emissions in Each Sector

The total amount of carbon emissions in 1949 was 0.66 Mt-$CO_2$, which increased to 6.02 Mt-$CO_2$, 8.51 Mt-$CO_2$, 13.93 Mt-$CO_2$, 29.64 Mt-$CO_2$, 54.05 Mt-$CO_2$ and 60.95 Mt-$CO_2$ in 1979, 1996, 2003, 2008, 2013 and 2016, respectively. From the comparison of the carbon emissions of each sub-model, it can be seen that the carbon emissions of the residential and commercial sectors were relatively small and relatively stable both before and after 1996. The carbon emissions of the two sectors showed a gradual upward trend. The increase in the commercial sector was greater than that of the residential sector, but not significant compared to the other sectors. Before 2003, the carbon emissions of the industrial sector showed a slow upward trend, but not as important as before 1979. During the short five years from 2003 to 2008, industrial carbon emissions showed a steady rise with a slope of more than 0.6. After 2008, the trend tended to be flat, but the rising slope was still close to 0.45. Before 1992, the carbon emissions of the transportation sector had similar trends and the same total emissions as the residential and commercial sectors. With the adjustment of the transportation structure from 1996 to 2003, the total amount of carbon emissions of the transportation sector fluctuated, but tended to be stable in 2008. After 2008, the carbon emissions of the transportation sector rose in a straight line. The carbon sink of green spaces changed shortly before 2003, and decreased greatly from 2003 to 2016, as seen in Figure 6.

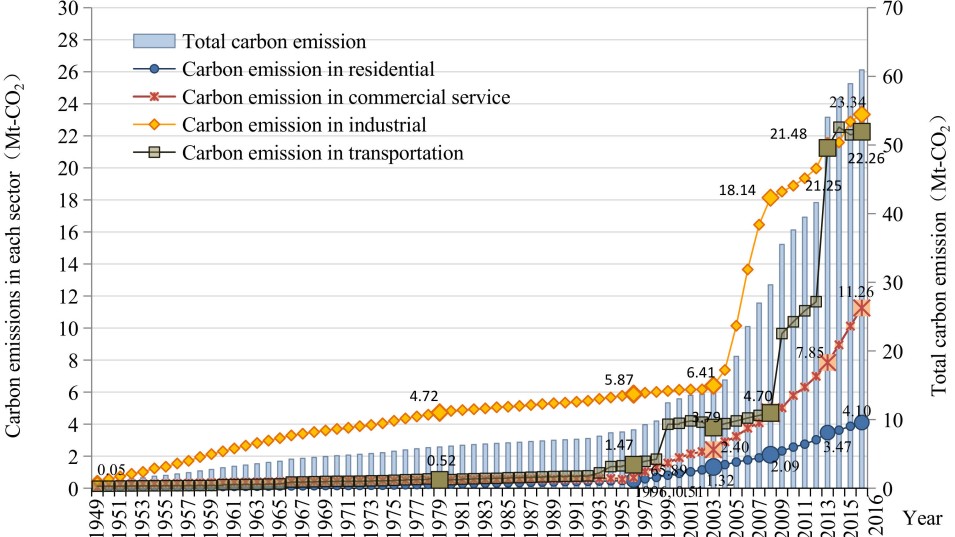

**Figure 6.** Total carbon emissions and emissions in each sector from 1949 to 2016.

### 4.2. The Contribution of Each Sector

Further statistical analysis was conducted on the sources of the increase of the total urban carbon emissions, which is the contribution of each urban subsystem to the total emissions. As shown in Figure 4, the proportion of carbon emissions of the residential sector before 1996 accounted for less than 6%, and the peak period of this proportion appeared from 2000–2008 and was between 6% and 10%. After 2005, the proportion began to decrease and went down to 6.73% in 2016. The proportion of the commercial sector decreased year by year, from 24.7% in 1950 to 6.8% in 1995, then increased gradually and kept floating between 10% and 20%. The industrial sector contributes the most to the total carbon emissions. Before 1996, the contribution rate of carbon emissions had been maintained at about 80%. From 1996 to 2000, the contribution rate of emissions decreased to about 50%. This state had been maintained until 2005. During 2006–2008, there was a new rise of 10% compared with the previous stage. After 2008, the industrial sector's contribution has continued to fall, and by the end of 2016, it had dropped from 50% to about 40%. The proportion of carbon emissions from transportation fluctuates greatly and is relatively unstable. Before 1986, the proportion was less than 10%, and it rose sharply from 1987 to 1996. In 2003, the transportation sector accounted for 27.2% of the total emissions. In 2008, it fell to 15.9%, and then rose directly to almost 40% of total emissions before slowly declining after 2013, as shown in Figure 7. The increase of traffic carbon emissions is mainly due to the traffic carbon emissions in the city, and the sum of passenger and freight carbon emissions accounts for only about 10% of the carbon emissions in this field. The increase of carbon emissions is linearly related to the ownership of cars, as shown in Figure 8.

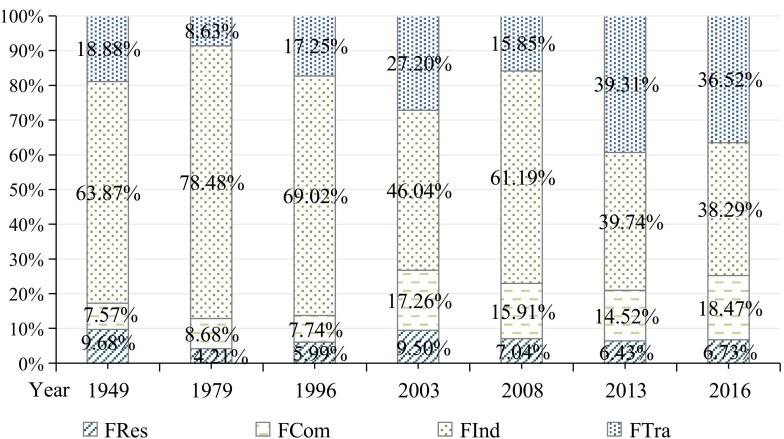

**Figure 7.** The structure of carbon emissions by sector from 1949 to 2016.

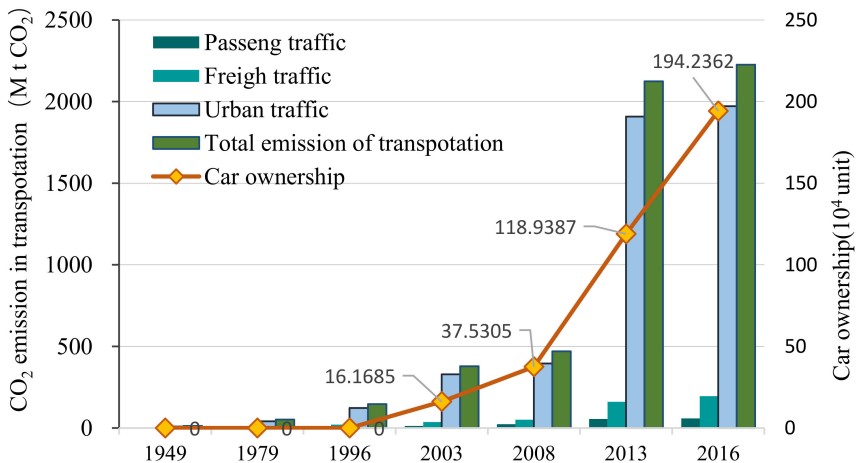

**Figure 8.** Growth trends of various transportation carbon emissions and car ownership changes.

### 4.3. Emission Intensity and per Capita Emissions

The emissions intensity was 186.11 t-$CO_2$/$10^4$CNY, 123.46 t-$CO_2$/$10^4$CNY, 6.58 t-$CO_2$/$10^4$CNY, 4.25 t-$CO_2$/$10^4$CNY, 2.51 t-$CO_2$/$10^4$CNY, 1.83 t-$CO_2$/$10^4$CNY and 1.33 t-$CO_2$/$10^4$CNY, in 1979, 1996, 2003, 2008, 2013 and 2016, respectively. The initial value of carbon emissions intensity was relatively high. In the following years, the carbon emissions intensity still increased and reached the peak value of 338.76 t-$CO_2$/$10^4$CNY in 1960. From 1960 to 1996, the carbon emissions intensity showed a linear downward trend, with an average annual decrease of 9.21 points. After 2006, the carbon emissions intensity still trended downward, with a decrease of 0.14 points per year; the per capita carbon emissions were 1.73 t-$CO_2$/104 people, 5.88 t-$CO_2$/$10^4$ people, 5.19 t-$CO_2$/$10^4$ people, 6.96 t-$CO_2$/$10^4$ people, 12.32 t-$CO_2$/$10^4$ people, 17.88 t-$CO_2$/$10^4$ people and 18.13 t-$CO_2$/$10^4$ people, in 1979, 1996, 2003, 2008, 2013 and 2016, respectively. From 1949 to 1966, the per capita carbon emissions increased slowly. During the 30 years from 1966 to 1996, the per capita carbon emissions remained stable and was basically maintained between 6.5 and 5.0, and after 2000, the per capita carbon emissions showed a stepwise upward trend (Figure 9).

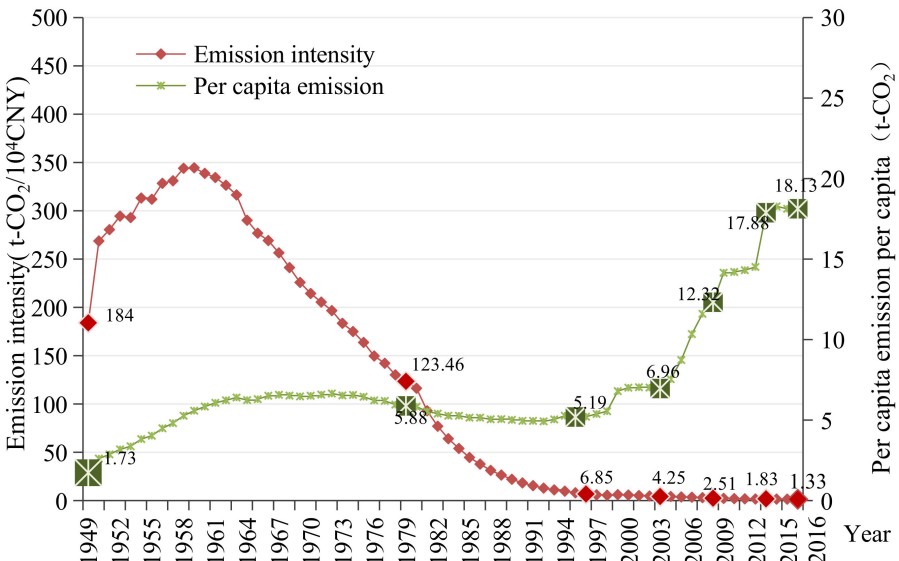

**Figure 9.** Emissions intensity and per capita emissions from 1949 to 2016.

### 4.4. Carbon Emissions per Unit Land Use

In terms of carbon emissions per unit of land use (Figure 10), before 1960, the trend line showed a slow upward trend, indicating that the carbon emissions per unit of land was increasing continuously. During 1960–1996, the trend line showed a year-on-year downward trend, with an obvious early downward trend, which gradually tended to be gentle later in this period. After 1996, there were some fluctuations in the line, and the general trend was rising. In order to analyze the carbon emissions intensity/carbon sink intensity of various types of land more clearly, the simulation was carried out in different sectors. It showed that the carbon emissions intensity of the residential and commercial sectors was relatively stable, and the trend line has always maintained a horizontal state of around 10 Mt-$CO_2$/ha. The carbon emissions intensity of the industrial sector was on an upward trend before 1960 and continued to decrease after 1960. The downward trend continued until 2003.

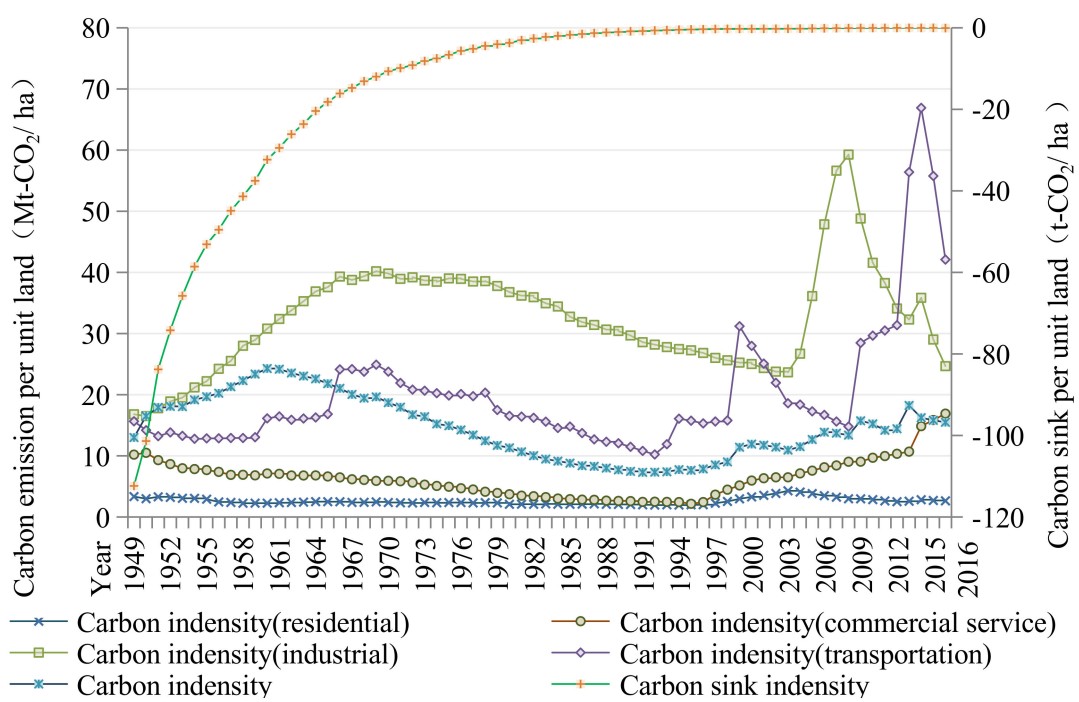

**Figure 10.** Carbon emission and carbon sink intensity per unit of land use.

*4.5. Correlation between Urban Spatial Evolution and Carbon Emissions*

The compactness of Changsha's urban form maintained a downward trend before 2008. Due to the construction of a large number of industrial parks in Changsha in 2008, the compactness of Changsha's urban form increased in 2008 and 2013, but the carbon emissions intensity has maintained a stable downward trend. There is a certain linear correlation between the compactness of urban form and the overall trend line of carbon emissions intensity, but the correlation is not absolute. The change of Changsha City's shape index is irregular, so it has no linear correlation with the growth rate of carbon emissions, as shown in Figure 11.

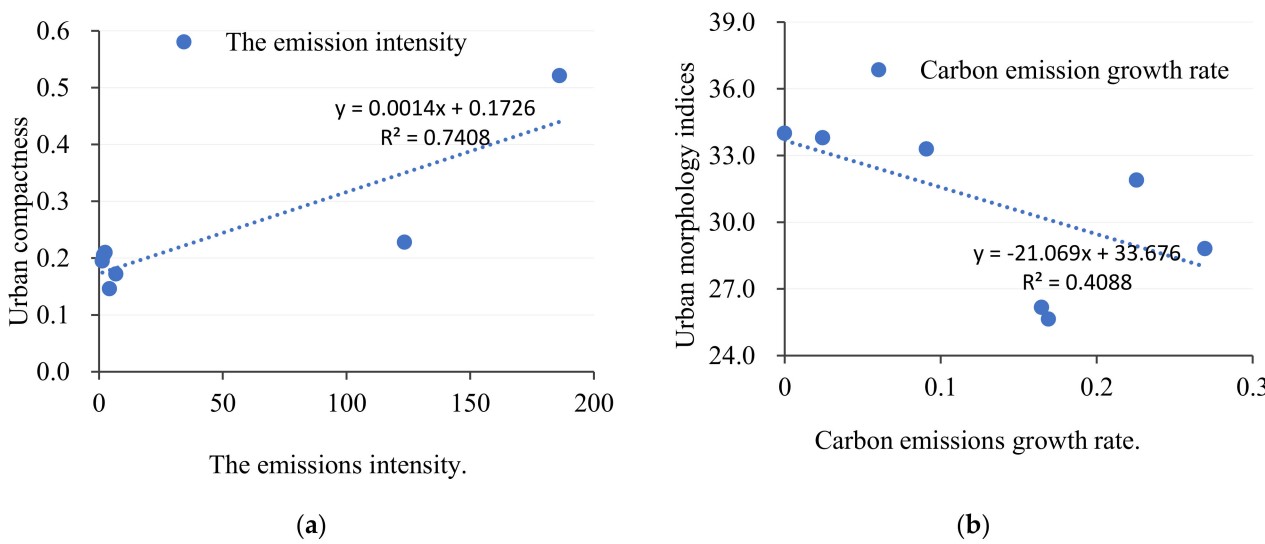

(**a**)            (**b**)

**Figure 11.** Correlation analysis. (**a**) Correlation analysis of the emissions intensity and urban compactness. (**b**) Correlation analysis of the carbon emissions growth rate and the urban morphology indices.

The results are shown in Table 3. The ranking of the relational grade for total carbon emissions and the land use area dedicated to each type of land is as follows: public

green space (G) > residential land (R) > road and transportation land (S) > industrial land (M) > commercial service land (A and B). The intensity of carbon emissions has a relatively large correlation with the intensity of commercial and service land expansion, followed by public green space, which has little correlation with the intensity of industrial land expansion.

**Table 3.** Relational grades of carbon emissions with land use area and urban sprawl intensity.

| Correlation Index | R | A&B | M | S | G |
|---|---|---|---|---|---|
| Carbon emissions and land use area | 0.6708 | 0.5358 | 0.5695 | 0.6274 | 0.7463 |
| The emissions intensity and urban sprawl intensity | 0.4619 | 0.6085 | 0.3866 | 0.3632 | 0.5521 |

## 5. Discussion

- The total carbon emissions and rate of $CO_2$ emissions per capita increase year by year. The per capita carbon emissions total was 6.96 t-$CO_2$ per capita, which was lower than the national average before 2008 (7.2 t-$CO_2$ in 2013), and became higher after 2008. In terms of emissions intensity, the carbon emissions per unit of GDP has decreased steadily from 344 t-$CO_2$/CNY10$^4$ (1958) to 1.33 t-$CO_2$/CNY10$^4$ (2016), while the carbon emissions per unit of land has generally declined, being stable in recent years only with a small fluctuation.

- There is a certain linear correlation between the compactness of urban form and the overall trend line of carbon emissions intensity, but the correlation is not absolute. The carbon emissions intensity has the greatest correlation with the expansion intensity of commercial land, followed by the expansion intensity of public green land, and has little correlation with the expansion intensity of industrial land.

- In terms of carbon emissions structure, the proportion of carbon emissions in the industrial sector is still important and accounts for up to 78%. Although, after 2008, with the increase in the intensity of industrial land expansion, the proportion of carbon emissions in the industrial sector had declined. However, the contribution rate is still close to 40%.

- Transportation carbon emissions have increased sharply with the increase of car ownership, accounting for more than 20% of the total emissions. Although Changsha rail transit began to operate in 2013, the growth trend of traffic carbon emissions has not been reduced. It shows that changes in the transportation structure cannot bring about rapid carbon reduction benefits.

- Nevertheless, the city is a huge system engineering. When using system dynamics for dynamic modeling, it is impossible to take all influencing factors into account. The urban form, patterns and neighborhood design issues (e.g., with respect to the same population density patterns, $CO_2$ emissions can result quite differently depending on the urban building typology and assembly) are also important factors affecting carbon emissions. In order to avoid the complexity of the model, the carbon emissions system dynamics model established in this paper is mainly applicable to the urban scale, rather than the block scale, and pays more attention to the urban materials space. Therefore, the model considers the relationship between urban population scale, land scale, economic growth and other factors, and energy consumption and carbon emissions. The three trends themselves are complex. Due to the lack of urban energy consumption statistics in small and medium-sized cities in China, the model is only applicable to large cities or mega cities with urban energy consumption statistics.

## 6. Conclusions

The dynamic model of the carbon emissions system is established by using the STELLA software platform, which can effectively evaluate the energy consumption and carbon emissions effects at the urban level. In this paper, Changsha City is selected as the case

study object, and the evaluation results can scientifically guide the formulation of urban carbon peak and carbon neutralization goals. The key findings are as follows:

- There may be a very important possible connection between carbon emissions and urban size. it is a remarkable issue, particularly in the southeastern city region settlements.
- It is necessary to strictly control the scale of cities, compact the urban spatial structure, and change the urban development model from "incremental development" to "stock development" to achieve smart urban growth. This move can promote the city of Changsha to achieve the 2030 carbon peak development goal as soon as possible.
- There is a need to speed up the adjustment and the optimization of the industrial and energy structures. Industrial development projects with low energy consumption and low emissions should be selected to settle in the industrial park so as to improve the energy use efficiency of industrial land. This is conducive to achieving the national policy goal of peaking carbon emissions in 2030 as soon as possible.
- Only by fundamentally reducing people's dependence on cars for travel and reducing the number of motor vehicles can real carbon reduction be achieved. Therefore, future plans should fit more public transport into the land use, encourage mixed land development, and focus on the coordination of transportation planning and land use planning. With the convenience of transportation, people will naturally choose public transportation to travel, further advocating that green transportation cannot be forced by the government.

**Author Contributions:** Conceptualization and methodology, L.L. and Y.T.; software, L.L. and K.B.B.; writing—original draft preparation, X.Z.; writing—review and editing, Y.C.; supervision, K.B.B. All authors have read and agreed to the published version of the manuscript.

**Funding:** This research was funded by the National Natural Science Foundation of China, grant number 31901363, the Key Forestry Majors Identified by State Forestry Administration, grant number [2016] NO. 21, the cultivation of "double first-class" disciplines in Hunan Province, grant number [2018] NO. 469, and the Natural Science Foundation of Hunan Province, China, grant number 2021JJ31156.

**Institutional Review Board Statement:** Not applicable.

**Informed Consent Statement:** Not applicable.

**Data Availability Statement:** The data are contained within the article. The data presented in this study are available on request from the corresponding author.

**Conflicts of Interest:** The authors declare no conflict of interest.

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
