# Peer review of "Urban Sprawl and Carbon Emissions Effects in City Areas Based on System Dynamics: A Case Study of Changsha City"

_applsci, doi:10.3390/app12073244_

Round 1
Reviewer 1 Report
A carefully prepared paper. Still some methodology issues are missing - one of them is why the city of Changsha was chosen? Does it have some particular outstanding parameters, or possibly the parameters are typical with other cities of its size.
I am also missing a discussion section. Conclusions do not include any limitations or contraines which can be found in this method.
Author Response
A carefully prepared paper. Still some methodology issues are missing - one of them is why the city of Changsha was chosen? Does it have some particular outstanding parameters, or possibly the parameters are typical with other cities of its size.
Response:
Thank you very much for your question. Supplementary reasons are given for choosing Changsha for the case study. Basically three reasons have motivated this choice, which are the importance of the city in terms of scale and rapid growth, making it worth paying attention to, then the completeness of the necessary data, which is hardly the case for many cities, and third but also indispensable the full accessibility of the research team to non-open-source data like urban land use data. The added paragraphs are shown below.
(P3 L130-135) Located in central China, Changsha is defined as an important city in the central region in the national development strategy. By July 2021, the population scale of Changsha's central urban area has exceeded 5 million, which is one of 17 mega cities in China (the population scale of central urban area is between 5-10 million, which is a mega city). Like other metropolises in developing countries, Changsha is undergoing rapid development and transformation, with an accelerated industrialization and urbanization, from 1949 to 2016, the urban population of Changsha has increased from 383,500 to 3.36 million, the regional GDP from 2.87 million CNY to 8510.13 million CNY, and the urban land use area from 6.7km2 to 476.34km2....
(P4 L156-161) First of all, Changsha is one of the most representative cities that is experiencing rapid urban expansion and economic development, and the carbon emissions of energy consumption are increasing. Secondly, the data on energy consumption over the years in the urban statistical Yearbook is relatively complete. Third, we have six consecutive urban land use data provided by Changsha Survey and Design Institute. Based on the above reasons, we selected Changsha City as the case study area.
I am also missing a discussion section. Conclusions do not include any limitations or contraines which can be found in this method.
Response:
Thank you so much for your suggestion. The major limitations and application scope of the model have been added in the Part5: P16 L505-518.
(P16 L505-518) Nevertheless, city is a huge system engineering. When using system dynamics for dynamic modeling, it is impossible to take all influencing factors into account. Although the urban form, patterns and neighborhood design issues (eg. relating to the same population density patterns, CO2 emissions can result quite differently depending on the urban building typology and assembly.) are also important factors affecting carbon emission. In order to avoid the complexity of the model, the carbon emission system dynamics model established in this paper is mainly applicable to the urban scale rather than the block scale, and pays more attention to the urban material space. Therefore, the model considers the relationship between urban population scale, land scale, economic growth and other factors and energy consumption and carbon emission. The three trends themselves are complex. Due to the lack of urban energy consumption statistics in small and medium-sized cities in China, the model is only applicable to large cities or mega cities with urban energy consumption statistics.
Reviewer 2 Report
The paper analyzes relationship between the mutual interactions and feedbacks between urban population, land expansion, economic growth, energy structure and carbon emission. this paper shows a dynamic model considering five urban operation subsystems: residential, commercial, industrial, transportation and green space carbon sink.
The topic is interest and relevant to the journal’s scope. However, major revisions are required for publication.
Abstract:
Lines 16-17: The quantitative assessment of urban energy consumption, carbon emission trends, and their emission effects is a prerequisite for achieving this goal.
Concordance in the verb.
Units need explanation: t-CO2/104CNY. What does CNY stand for?
Introduction:
Lines 61 and 63. There are acronyms that are not explained, for example SOC and LMDI.
LMDI is explained in line 91: Log mean Divisia index method.
Please, provide a list of acronyms and units.
Figure 1: The legend font size is too small. Is the figure original or has it been taken from another source?
The text explaining equations (1) (2) and (3) refers to “I” (upper case), but in the formula there is “i” (lower case). Please clarify.
Figure 3. The text refers to Figure 3 (a), (b), (c) in section 3.5. However, in section 3.5. Transportation Sub-Model, the text keeps referring Figure 3(d) and (e). I suggest to split Figure 3 into two different figures and place them in two different subsections.
The section Discussion does not reflect the great amount of data that have been presented in Results sections.
The sentence: “Transportation carbon emissions have increased sharply with the increase of car ownership, accounting for more than 20% of the total emission” is an example of meaningful conclusions. Please, add more data analysis and trends in the conclusions section.
Can you include any limitations you have found in the method and some future research lines?
Author Response
The paper analyzes relationship between the mutual interactions and feedbacks between urban population, land expansion, economic growth, energy structure and carbon emission. this paper shows a dynamic model considering five urban operation subsystems: residential, commercial, industrial, transportation and green space carbon sink.
The topic is interest and relevant to the journal’s scope. However, major revisions are required for publication.
Abstract:
Lines 16-17: The quantitative assessment of urban energy consumption, carbon emission trends, and their emission effects is a prerequisite for achieving this goal.
Concordance in the verb.
Response:
Thank you so much for your suggestion. The verb “is” was in concordance with “The quantitative assessment” as the subject. But the sentence has been rewritten as follow, for more clarity:
(P1 L16-17) The quantitative evaluation of the trend of urban energy consumption and carbon emission trends is a premise for achieving this goal.
Units need explanation: t-CO2/104CNY. What does CNY stand for?
Response:
Thank you very much. CNY stands for the currency, ChinaYuan. It is now written before the numbers as in the international standard. From the corrected manuscript you will read:
(P1 L31) From the perspective of carbon emission effects, carbon emissions per unit of GDP had a clear downward trend, from 186.11t-CO2/CNY104 to 1.33t-CO2/CNY104;...
Introduction:
Lines 61 and 63. There are acronyms that are not explained, for example SOC and LMDI.
LMDI is explained in line 91: Log mean Divisia index method.
Please, provide a list of acronyms and units.
Response:
Thank you so much for your advice. The two acronyms, SOC and LMDI, are now explained. In line 63 and line 65, you will read:
(P2 L65 and L67) Kabindra Adhikari propose a space-for-time substitution model to predict current and future soil organic carbon (SOC) stocks in Wisconsi [8]. At the regional level, Liu,Y. took the carbon emission panel data of six provinces in Central China as the basic data, and used the LMDI decomposition method to study the temporal and spatial evolution characteristics and convergence of carbon emission [9].
Figure 1: The legend font size is too small. Is the figure original or has it been taken from another source?
Response:
Thank you so much for your question. Yes, indeed, the figure is original; the legend font has been enlarged (see Figure 1).
The text explaining equations (1) (2) and (3) refers to “I” (upper case), but in the formula there is “i” (lower case). Please clarify.
Response:
Thank you for your reminder. Sorry, this was an error. We have changed the "I" (upper case) into to i (lowercase).
(P6 L246) ..., i is the energy type, which refers to coal, liquefied petroleum gas (LPG), gas, natural gas (NG) or electricity in the residential sector.
(P7 L268) ...,i is the energy type, which refers to liquefied petroleum gas (LPG), gas, natural gas (NG) or electricity in the commercial service sector.
(P8 L287) ..., i is the energy type, which refers to raw coal (F1), washed coal (F2), other washed coal (F3), coal products (F4), coke (F5), producer gas (F6), NG (F7), liquefied natural gas (F8), gasoline(F9), kerosene (F10), diesel (F11), fuel oil (F12), LPG (F13), other petroleum products (F14), heat (F15), electricity(F16) and other biofuels (F17) in industrial sector.
Figure 3. The text refers to Figure 3 (a), (b), (c) in section 3.5. However, in section 3.5. Transportation Sub-Model, the text keeps referring Figure 3(d) and (e). I suggest to split Figure 3 into two different figures and place them in two different subsections.
Response:
Yes, that is a good suggestion. Accordingly, the figures have been split and put after the relevant paragraphs for a better readability. (see Figure 3-5).
The section Discussion does not reflect the great amount of data that have been presented in Results sections.
The sentence: “Transportation carbon emissions have increased sharply with the increase of car ownership, accounting for more than 20% of the total emission” is an example of meaningful conclusions. Please, add more data analysis and trends in the conclusions section.
Response:
Thank you so much for your advice. More analysis details are now added to the discussions, now under the Conclusion section.
(P15 L459-517) This paper establishes a dynamic model, aiming at assessing the energy consumption and the carbon emission effect at the urban level, using STELLA software. Focusing on the case of Changsha, strong measures were taken to continuously improve energy efficiency around the goal of carbon peak and carbon neutrality. The conclusions and inspirations are as follows:
- The total carbon emission and and rate of CO2emission per capita increase year by year. The per capita carbon emission was 6.96t-CO2per capita, lower than the national average before 2008 (7.2 t-CO2 in 2013) and became higher after 2008. There may be a very important possible connection between carbon emissions and urban size. it is a remarkable issue particularly in the south-eastern city regions settlements. In terms of emission intensity, the carbon emission per unit of GDP has decreased steadily from 344t-CO2/CNY104 (1958) to 1.33t-CO2/CNY104 (2016), while the carbon emission per unit of land has generally declined being stable in recent years only a little small fluctuation.
- There is a certain linear correlation between the compactness of urban form and the overall trend line of carbon emission intensity, but the correlation is not absolute. The carbon emission intensity has the greatest correlation with the expansion intensity of commercial land, followed by the expansion intensity of public green land, and has little correlation with the expansion intensity of industrial land. Therefore, it is necessary to strictly control the scale of cities, compact the urban spatial structure, and change the urban development model from "incremental development" to "stock development" to achieve smart urban growth. This move can promote the city of Changsha to achieve the 2030 carbon peak development goal as soon as possible.
- In terms of carbon emission structure, the proportion of carbon emission in the industrial sector is still important, accounting for up to 78%. Although after 2008, with the increase in the intensity of industrial land expansion, the proportion of carbon emission in the industrial sector has declined. However, the contribution rate is still close to 40%. There is a need to speed up the adjustment and the optimization of the industrial and energy structures. Select industrial development projects with low energy consumption and low emission to settle in the industrial park, so as to improve the energy use efficiency of industrial land. It is conducive to achieving the national policy goal of peaking carbon emissions in 2030 as soon as possible.
- Transportation carbon emissions have increased sharply with the increase of car ownership, accounting for more than 20% of the total emission. Although Changsha rail transit began to operate in 2013, the growth trend of traffic carbon emissions has not been reduced. It shows that changes in the transportation structure cannot bring about rapid carbon reduction benefits. Only by fundamentally reducing people's dependence on cars for travel and reducing the number of motor vehicles can real carbon reduction be achieved. Therefore, the future plans should fit more public transport into the land use, encourage mixed land development, and focus on the coordination of transportation planning and land use planning. With the convenience of transportation, people will naturally choose public transportation to travel, advocating that green transportation cannot be forced by the government.
- Nevertheless, city is a huge system engineering. When using system dynamics for dynamic modeling, it is impossible to take all influencing factors into account. Although the urban form, patterns and neighborhood design issues (eg. relating to the same population density patterns, CO2emissions can result quite differently depending on the urban building typology and assembly.) are also important factors affecting carbon emission. In order to avoid the complexity of the model, the carbon emission system dynamics model established in this paper is mainly applicable to the urban scale rather than the block scale, and pays more attention to the urban material space. Therefore, the model considers the relationship between urban population scale, land scale, economic growth and other factors and energy consumption and carbon emission. The three trends themselves are complex. Due to the lack of urban energy consumption statistics in small and medium-sized cities in China, the model is only applicable to large cities or mega cities with urban energy consumption statistics.
Can you include any limitations you have found in the method and some future research lines?
Response:
Thank you so much for your suggestion. The discussion (now conclusion) section have been modified according to your suggestion, and the main limitations of this research have been specified.
Reviewer 3 Report
The article properly account for a recent work carried out by applying a specific methods to appraise and foreseen C02 equivalent emission patterns under different urban development functional regime and future scenarios relating to an urban China case.
The structure of the article turns out being meaningfully clear whereas the goals, methodology and results are properly and clearly presented. That in such a way to shed light on some relevant matters suitable to address urban policies, especially relating to effectiveness of the joint adoption of technological and spatial planning solutions.
Nevertheless, relating to that and to better tap into the interesting results provided by the study, the discussion part, once renamed as discussion/conclusion section, could be slightly improved especially focusing, in reflexive form, to:
- Discuss the Limits of the analysis model adopted. That especially considering how the urban form, patterns and neighborhood design issues are not taken in account by the model, whereas issues of morphology strongly affect energy consumption especially relating to residentials sectors in terms of building typology functionings and energy performances (including embodied energy) and inducted citizens behaviours. (e.g. relating to the same population density patterns, CO2 emissions can result quite differently depending on the urban building typology and assemblage) (see among others Rickwood , 2009).
- Take in account typology of land use aspects (e.g less or more mixed form) that are greatly considered in existing literature and inductive of different patterns and mode of mobility.
- Mention the matter of the possible connection between rate of CO2 (or GHGs equivalent) emission per capita and urban size , a remarkable issue particularly in the south-eastern city regions settlements.
Moreover, related to the latter point, the trend in the relative composition of CO2 emission for sub-sector, with the growing role of transportation -along with the meaningful rates of energy intensity in the industrial and residential sector pointed out by the article– calls for some some reflections about the role of the settlement patterns and urban dimension to be pursued in policies, beyond functional and technological remedies. (e.c. monocentric city region models Vs urban containment and polycentric patterns) to cope with carbon neutral development.
The authors quotations in the text are missing year;
References to literature about the matter are quite limited, especially once considered the issues raised by this comment.
Author Response
The article properly account for a recent work carried out by applying a specific methods to appraise and foreseen C02 equivalent emission patterns under different urban development functional regime and future scenarios relating to an urban China case.
The structure of the article turns out being meaningfully clear whereas the goals, methodology and results are properly and clearly presented. That in such a way to shed light on some relevant matters suitable to address urban policies, especially relating to effectiveness of the joint adoption of technological and spatial planning solutions.
Nevertheless, relating to that and to better tap into the interesting results provided by the study, the discussion part, once renamed as discussion/conclusion section, could be slightly improved especially focusing, in reflexive form, to:
Discuss the Limits of the analysis model adopted. That especially considering how the urban form, patterns and neighborhood design issues are not taken in account by the model, whereas issues of morphology strongly affect energy consumption especially relating to residentials sectors in terms of building typology functionings and energy performances (including embodied energy) and inducted citizens behaviours. (e.g. relating to the same population density patterns, CO2 emissions can result quite differently depending on the urban building typology and assemblage) (see among others Rickwood , 2009).Take in account typology of land use aspects (e.g less or more mixed form) that are greatly considered in existing literature and inductive of different patterns and mode of mobility.
Mention the matter of the possible connection between rate of CO2 (or GHGs equivalent) emission per capita and urban size , a remarkable issue particularly in the south-eastern city regions settlements.
Moreover, related to the latter point, the trend in the relative composition of CO2 emission for sub-sector, with the growing role of transportation -along with the meaningful rates of energy intensity in the industrial and residential sector pointed out by the article–calls for some some reflections about the role of the settlement patterns and urban dimension to be pursued in policies, beyond functional and technological remedies. (e.c. monocentric city region models Vs urban containment and polycentric patterns) to cope with carbon neutral development.
Response:
Thank you so much for your suggestion. The additions made to the discussion (now conclusion) section have taken into account these three points.
The authors quotations in the text are missing year;
Response:
Thank you so much for your suggestion. The authors quotations have been revised according to the journal template.
References to literature about the matter are quite limited, especially once considered the issues raised by this comment.
Response:
Thank you so much for your suggestion. Throughout the revised manuscript, further details and relevant literatures have been added. For instance, the two paragraphs below provide more details about connections between carbon emmisions and urban size
(P2 L72-97) Imran Hanif employs the Environmental Kuznets Curve (EKC) hypothesis in studying the impact of energy consumption, economic growth, and urbanization on carbon emissions in developing economies[11].......Wu,M. modeled and analyzed the carbon emission effect of land use under different policy scenarios in Wuhan from the impact of land, population, society, economy and energy on carbon emission [17]. Xu,J. modeled from the perspective of landscape ecological risk and environmental pollution, and compared the landscape risk assessment of four urban layout modes [18]. Sun,Y.H. constructed an urban sustainable development evaluation model covering five subsystems: economy, population, water resources, environment, science and technology and education [19].
A GIS-based model was proposed for testing how urban form and building typology affect energy performance and carbon emissions in the City of Macau, ref.[20]. Carpio,A. analyses the urban expansion of the Monterrey Metropolitan Area (MMA) Mexico from 1990 to 2019 using satellite imagery and Geographic Information Systems (GIS) to determine its relation to carbon emission[21]. With figures showing that approximately 40% of the UK's carbon emissions are attributable to household and transport behaviour, policy initiatives have progressively focused on the facilitation of "sustainable behaviours", ref. [22].
Reviewer 4 Report
I think that the authors could remove the term "spatial" from the title and throughout the paper. Simply "urban sprawl" will be enough. The term is already widely known, especially for its spatial pattern.
The Abstract is generally well written in terms of the overall necessity of the addressed topic. However, there is no clear connection made between urban sprawl and carbon emissions.
The Introduction largely focuses on emissions related issues. The relationship with urbanization and urban sprawl should be also highlighted. There are almost no references to urban sprawl, and it influences emissions.
Given that the topic is significant for the wide scientific community and practice, the paper needs the use of a broader scientific background on the issues at stake. The literature is relatively poor and insufficient.
Lines 39-41. Please reformulate the phrase, it is rather unclear.
Line 44 the rise of air temperature increases inside cities, as well. Not only around cities.
Lines 51, 53, 56 and more, the way some authors are citated in the text (e.g., Genice, Flavio, Geoffrey) should be reconsidered.
Line 85 - What do you mean by "new towns"? Please clarify.
Line 19. The authors mentioned Figure 1 and Table 1 without even a short analysis of the land use dynamics and its relationship with urban sprawl.
Line 28. Land use dynamics would be more appropriate than "present land use" since you are referring to various years.
Table 1. The caption of the table is "urban land use area" and the table head mentions "land use types". So, the authors are referring to land use in general or urban land use?
Is urban land use the same with built-up area? It is not clear. The table includes several land use types. What is development land? Please use CLC nomenclature for accuracy.
It would be useful to have either a separate figure with the location of the study area in China, or a small map attached to Figure 1.
The Results are too technical, they should be better linked with the underlying factors. The link between the emissions and urban sprawl should be strengthened.
Lines 55-57 please reformulate; there is repeatability and misuse of words.
The Discussions should focus on analysing the current findings in relation to what was published so far in the literature. The current Discussions looks more like Conclusions.
Conclusions are missing.
Author Response
I think that the authors could remove the term "spatial" from the title and throughout the paper. Simply "urban sprawl" will be enough. The term is already widely known, especially for its spatial pattern.
Response:
Thank you so much for your suggestion. We have removed the term "spatial" from the title considering your suggestions.
The Abstract is generally well written in terms of the overall necessity of the addressed topic. However, there is no clear connection made between urban sprawl and carbon emissions.
Response:
Thank you so much. In the abstract section, the relationship of urban sprawl to carbon emissions are now mentioned in the abstract.
(P1 L33-36) There is a certain linear correlation between the compactness of urban shape and the overall trend of carbon emission intensity, while the urban shape index has no linear correlation with the growth rate of carbon emissions.
The Introduction largely focuses on emissions related issues. The relationship with urbanization and urban sprawl should be also highlighted. There are almost no references to urban sprawl, and it influences emissions.
Given that the topic is significant for the wide scientific community and practice, the paper needs the use of a broader scientific background on the issues at stake. The literature is relatively poor and insufficient.
Response:
Thank you so much for your suggestion. More references have been added regarding the relation of urban sprawl and carbon emissions. Precisions are added as in the following paragraph.
(P2 L72-97) Imran Hanif employs the Environmental Kuznets Curve (EKC) hypothesis in studying the impact of energy consumption, economic growth, and urbanization on carbon emissions in developing economies[11]. ......A GIS-based model was proposed for testing how urban form and building typology affect energy performance and carbon emissions in the City of Macau, ref.[20]. Carpio,A. analyses the urban expansion of the Monterrey Metropolitan Area (MMA) Mexico from 1990 to 2019 using satellite imagery and Geographic Information Systems (GIS) to determine its relation to carbon emission[21]. With figures showing that approximately 40% of the UK's carbon emissions are attributable to household and transport behaviour, policy initiatives have progressively focused on the facilitation of "sustainable behaviours", ref. [22].
Lines 39-41. Please reformulate the phrase, it is rather unclear.
Response:
Thank you so much for your suggestion. This phrase (Lines 39-41) has been reformulate according to your remark.
(P1 L42-44) Climate change is a global problem facing mankind. "Peak CO2 emissions by 2030 and achieve carbon neutrality by 2060." This is China's strong commitment to addressing global climate change.
Line 44 the rise of air temperature increases inside cities, as well. Not only around cities.
Response:
Thank you so much for your suggestion. This phrase (L 44) has been reformulate according to your remark.
(P2 L47-48) Up to 80% of greenhouse gas (GHG) emissions are generated by more than half of the population living in urban areas [2]
Lines 51, 53, 56 and more, the way some authors are citated in the text (e.g., Genice, Flavio, Geoffrey) should be reconsidered.
Response:
Thank you so much for your suggestion. The format of author citation in the text has been revised as follow:
(P2 L53, 55, 58) Genice,K. using Mexico as an example, proposed eight energy efficiency measures with regard to energy use in residential, commercial and public sectors [3]. Flavio,R. describes the results of a study of Ecuador's energy status, using the system dynamics methodology to model supply, demand and CO2 emissions scenarios for the year 2030 [4].
Line 85 - What do you mean by "new towns"? Please clarify.
Response:
Thank you so much for your question. The concept of New Town in China generally refers to newly urban developments areas; this explanation has been added in P2 Lines 97-99
(P2 L97-99) The research at the urban level mainly focuses on the research of new towns. The concept of New Town in China generally refers to newly urban developments areas. .
Line 119. The authors mentioned Figure 1 and Table 1 without even a short analysis of the land use dynamics and its relationship with urban sprawl.
Response:
Thank you so much, this suggestion has been considered providing more analytical details as follow:
(P3 L140-148) Figure 1 and table 1 show that the urban construction land is increasing, while the agricultural and forestry land is decreasing, the urban compactness shows a downward trend, "spread out like a pie", and the urban shape index has little change. The expansion intensity of residential land is relatively the largest, especially after 2008, which is several times that of other types of land; The expansion intensity of commercial service land is relatively stable; The expansion intensity of industrial land continues to decline, and the expansion intensity of road traffic land is relatively stable.
Line 128. Land use dynamics would be more appropriate than "present land use" since you are referring to various years.
Response:
Yes, it is a good suggestion, corrections are made accordingly.
Table 1. The caption of the table is "urban land use area" and the table head mentions "land use types". So, the authors are referring to land use in general or urban land use?
Response:
Thank you so much for your question. The caption of the table has been reformulated and the content has been reorganized for a better understanding.
Table 1. Urban land use indices, urban compactness and urban morphology indice.
|
Item |
1949 |
1979 |
1996 |
2003 |
2008 |
2013 |
2016 |
|
|
The built-up area |
6.7 |
51.8 |
104.93 |
96.26 |
209.63 |
393.78 |
476.34 |
|
|
Urban land area |
Residential land(R) |
1.92 |
11.02 |
25.38 |
30.95 |
69.95 |
134.79 |
155.32 |
|
Commercial service land(A&B) |
1.26 |
13.33 |
26.97 |
37.02 |
43.98 |
73.4 |
66.58 |
|
|
Industrial land(M) |
1.52 |
12.5 |
21.9 |
27.1 |
30.6 |
66.5 |
94.56 |
|
|
Road and transportation land(S) |
0.8 |
2.97 |
9.59 |
20.58 |
31.76 |
37.67 |
52.87 |
|
|
Public green space(G) |
0.6 |
4.8 |
7.94 |
12.29 |
21.42 |
29.13 |
34.13 |
|
|
Waters (E1) |
25.26 |
73.45 |
82.3 |
85.6 |
85.6 |
93 |
95.65 |
|
|
Arable land (E21) |
193.89 |
177.83 |
151.73 |
143.61 |
104.68 |
73.5 |
52.3 |
|
|
Garden land (E22) |
179.38 |
164.53 |
140.38 |
132.87 |
96.84 |
68 |
35.2 |
|
|
Forest land (E23) |
96.02 |
88.07 |
75.14 |
71.12 |
51.84 |
36.4 |
24.3 |
|
|
Urban compactness (U) 1 |
0.521 |
0.228 |
0.172 |
0.146 |
0.21 |
0.205 |
0.195 |
|
|
Urban morphology indice (I)2 |
34.0 |
28.81 |
33.80 |
33.29 |
31.89 |
26.17 |
25.64 |
|
Note: urban land use area contains eight categories of R, A, B, M, W, S, U, and G, code for classification of urban land refers to GB50137-2011. W (warehouse land) and U (utility land) land have little to do with urban carbon emissions. Therefore, class W and U land are not described in the table. The unit area for each type of urban land is sq. km. 1 [25] proposed that the methodology for measuring compactness was adopted based on the urban land use GIS raster analysis and resorting to the gravitation approach. 2 The urban morphology indices are obtained by calculating the ratio of the standard circular shape to the urban boundary shape [26]. Urban morphology indices (I)=Σni=1|(Ri/Σni=1Ri×100-100/n)|, where r is the radial distance as measured from the centroid of the built-up area to its edge and ,n is the number of regularly spaced radii.
Is urban land use the same with built-up area? It is not clear. The table includes several land use types. What is development land? Please use CLC nomenclature for accuracy.
Response:
Thank you so much for your question. The built-up area was meant to express the total area constructed, as an indicator of Land use. But the table as revised should not pose such a confusion.
It would be useful to have either a separate figure with the location of the study area in China, or a small map attached to Figure 1.
Response:
Thank you so much for your question. Figure 1 has been revised considering this suggestions.
The Results are too technical, they should be better linked with the underlying factors. The link between the emissions and urban sprawl should be strengthened.
Response:
Thank you so much for your suggestion. In addition to adding urban sprawl indicators such as Urban morphology indice, urban compactness, etc In Table 1 and table 2 of Section 2.1. We have also added to the results section explanations about the relationship between urban expansion and carbon emissions. In addition, the discussion (now in the Conclusion section) was rewritten. The revised text strengthens the analysis on relationships between urban expansion and carbon emissions.
Lines 55-57 please reformulate; there is repeatability and misuse of words.
The Discussions should focus on analysing the current findings in relation to what was published so far in the literature. The current Discussions looks more like Conclusions.
Conclusions are missing.
Response:
Thank you so much for your suggestion. This was a mistake, in fact the content of the Discussion was meant for te Conclusion of the article. Therfore, that section has been renamed Conclusion.
Round 2
Reviewer 1 Report
The authors have reconsidered and responded to the remarks, I still think that methodology section could be better - but this can be done in the next paper.
Author Response
The authors have reconsidered and responded to the remarks, I still think that methodology section could be better - but this can be done in the next paper.
Response:
Thanks a lot. This remark will surely be taken into account.
Reviewer 2 Report
The authors have addressed all the previous comments.
The manuscript is accepted for publication after minor format changes.
Author Response
The authors have addressed all the previous comments.
The manuscript is accepted for publication after minor format changes.
Response:
The manuscript is revised as required. If need be, authors are open to further indications to ameliorate the manuscript’s format.
Reviewer 4 Report
The authors took into account most of the recommendations and suggestions and there is a general improvement of the manuscript. However, the authors have made very little progress in the expansion of the scientific literature on the subject and in expanding the relationship with urban sprawl.
Also, please pay attention to the appropriate wording (Lines 136 and 159). Land use dynamics means the evolution/dynamics between several years not for each year at a time.
I suggest for Figure 1 "Land use dynamics (1979, 1996, 2003, 2008, 2013 and 2016)". You don’t have to use the word "maps" in the figure caption, it is obvious that there are maps involved.
Author Response
The authors took into account most of the recommendations and suggestions and there is a general improvement of the manuscript. However, the authors have made very little progress in the expansion of the scientific literature on the subject and in expanding the relationship with urban sprawl.
Response:
Thank you so much for your suggestion. The literature review has been further improved with more references considered your previous suggestions.
(P3 L52-132) ....At the municipal level, mainly four aspects have been considered: low-carbon construction and urban development, carbon emission and urban spatial structure, carbon emission and urban form, carbon emission and urban sprawl. For instance, Pan,H.X. analyzes the methodology of urban planning formulation at region, city and street block level, and several arguments on urban land use, transport, density control and mixture of land use have been discussed towards low carbon urban spatial strategy [13]. ....Guo,J. points out that the key of urban development transformation is the coordination of the industrial structure and the optimization of the spatial structure to achieve low carbon goals [23]. A GIS-based model was proposed for testing how urban form and building typology affect energy performance and carbon emissions in the City of Macau, ref.[24]. Ewing,R. suggests that compact urban spaces can effectively reduce the energy consumption of the urban residents, and thus reduce the carbon dioxide emissions [25]. Chen, Z.Q. used the regression model to test the impact of urban form on carbon emissions, he also put forward a low carbon development framework of "spatial-land-transportation" to reduce the level of urban carbon emissions [26]. Carpio,A. analyses the urban expansion of the Monterrey Metropolitan Area (MMA) Mexico from 1990 to 2019 using satellite imagery and Geographic Information Systems (GIS) to determine its relation to carbon emission[27]. Hong,Y. indicates that designing cities to be compact with less accessibility to Green space (GS) and water body (WB) may increase household energy consumption. Substantial use of GS and WB, even if they are fragmented, will reduce carbon emission from residential energy use [28]. Li,J.J. analyzed the relationship between land use and carbon emission intensity in Shanghai. The study specifically established the relashionship betwen urban compactness and industrial land use using three models: a space compactness model, an energy consumption model and a carbon emission intensity model [29]. The experiments conducted by Shao,Z.F [30] showed that urban sprawl resulted in unsustainable urban development patterns from the social, environmental, and economic perspectives. Zhang,M. calculated the levels of land intensive use and land use carbon emission from 1996 to 2010 in 3 central cities in Hubei Province; the results show that there is a long-term equilibrium between intensive land use level and land use carbon emission[31].
Also, please pay attention to the appropriate wording (Lines 136 and 159). Land use dynamics means the evolution/dynamics between several years not for each year at a time.
I suggest for Figure 1 "Land use dynamics (1979, 1996, 2003, 2008, 2013 and 2016)". You don’t have to use the word "maps" in the figure caption, it is obvious that there are maps involved.
Response:
Thanks for this suggestion, the Figure caption has been corrected into “Land use change from 1979 to 2016”.